

# Changes in muscle activity during the flexion and extension phases of arm cycling as an effect of power output are muscle-specific

Carla P. Chaytor, Davis Forman, Jeannette Byrne, Angela Loucks-Atkinson and Kevin E. Power

Human Kinetics and Recreation, Memorial University of Newfoundland, St. John's, Newfoundland, Canada

## ABSTRACT

Arm cycling is commonly used in rehabilitation settings for individuals with motor impairments in an attempt to facilitate neural plasticity, potentially leading to enhanced motor function in the affected limb(s). Studies examining the neural control of arm cycling, however, typically cycle using a set cadence and power output. Given the importance of motor output intensity, typically represented by the amplitude of electromyographic (EMG) activity, on neural excitability, surprisingly little is known about how arm muscle activity is modulated using relative workloads. Thus, the objective of this study was to characterize arm muscle activity during arm cycling at different relative workloads. Participants ($n = 11$) first completed a 10-second maximal arm ergometry sprint to determine peak power output (PPO) followed by 11 randomized trials of 20-second arm cycling bouts ranging from 5–50% of PPO (5% increments) and a standard 25 W workload. All submaximal trials were completed at 60 rpm. Integrated EMG amplitude (iEMG) was assessed from the biceps brachii, brachioradialis, triceps brachii, flexor carpi radialis, extensor carpi radialis and anterior deltoid of the dominant arm. Arm cycling was separated into two phases, flexion and extension, relative to the elbow joint for all comparisons. As expected, iEMG amplitude increased during both phases of cycling for all muscles examined. With the exception of the triceps brachii and extensor carpi radialis, iEMG amplitudes differed between the flexion and extension phases. Finally, there was a linear relationship between iEMG amplitude and the %PPO for all muscles during both elbow flexion and extension.

## INTRODUCTION

Arm cycling, also referred to as arm crank ergometry, is commonly used as a means of exercise in rehabilitation programs for individuals living with upper and/or lower limb impairments following, for example, stroke or spinal cord injury. While the benefits of this type of exercise for the metabolic and cardiovascular systems are important, an additional aim when used in neurological populations is to maintain functional motor output or to induce neural plasticity (*Kaupp et al., 2018*; *Klarner et al., 2016*), potentially leading to a regain of motor output in the affected limb(s). Given the importance of arm

Corresponding author
Kevin E. Power, kevin.power@mun.ca

cycling to rehabilitation and the knowledge that exercise-induced adaptations are often intensity-dependent, surprisingly little information is available regarding how arm cycling intensity influences the activation of the arm musculature.

It is well-known that as muscle contraction intensity increases, so too will muscle activity as assessed via surface EMG, at least up to a certain intensity. Using the gastrocnemius muscle, Lippold was amongst the first to demonstrate that force and EMG increased linearly during isometric contractions (*Lippold, 1952*). Since that time, numerous studies have examined the EMG-force relationship in various muscles during isometric contractions, with examples of linear and non-linear relationships having been demonstrated (*Woods & Bigland-Ritchie, 1983*). Assessing the EMG-force relationship during dynamic muscle contractions is more challenging due to numerous physiological and non-physiological factors (*Farina, 2006*). Despite these issues, linear relationships between peak velocity and acceleration with the EMG amplitude of the elbow extensors during a ballistic elbow extension has been shown (*Aoki, Nagasaki & Nakamura, 1986*) with similar results reported in the elbow flexors (*Barnes, 1980*).

The isometric and dynamic motor outputs that have been examined, however, are fundamentally different than locomotor outputs (examples include leg cycling and arm cycling) which are characterized by the bilateral, rhythmic, and alternating activation of antagonistic motoneurone pools and are under different neural control than isometric contractions (*Carroll et al., 2006*; *Forman et al., 2014*; *Forman et al., 2016a*; *Forman et al., 2016b*; *Sidhu et al., 2012*; *Zehr & Duysens, 2004*). The activation of various muscles during leg cycling has been studied in detail, with the influence of factors such as pedalling rate and workload having been examined (for detailed review see Hug & Dorel 2009). As with the results obtained using isometric and other dynamic contractions, both linear (*Bigland-Ritchie & Woods, 1974*; *Duchateau, Le Bozec & Hainaut, 1986*; *Taylor & Bronks, 1994*) and non-linear relationships (*Duchateau, Le Bozec & Hainaut, 1986*; *Hug et al., 2006*; *Hug et al., 2003*; *Lucia et al., 1997*) between changes in power output and EMG have been demonstrated. Interestingly, by increasing the power output via increased load, Duchateau and colleagues showed a linear EMG-power output relationship with the soleus and a non-linear relationship with the gastrocnemius, suggesting that muscle-dependent differences may exist (*Duchateau, Le Bozec & Hainaut, 1986*).

These EMG relationships are further complicated between cycling phases. During arm cycling, data is frequently discussed in terms of the two main propulsion phases; the flexion phase, which pulls the handle towards the individual (driven by the elbow flexors), and the extension phase, which pushes the handle away (driven by the elbow extensors). *Smith et al. (2008)* was some of the first work to show that muscle activity of upper-limb muscles was continuously influenced throughout the crank cycle (see Figures 2A and 2B in *Smith et al. (2008)*). Recently, we demonstrated that there are unique phase differences between the biceps and triceps brachii during arm cycling (*Forman, Monks & Power, 2019*; *Forman et al., 2015*). While the biceps brachii exhibits cyclical muscle activity, with large bursts in the EMG signal during the flexion phase and almost no activity in the extension phase, the triceps brachii demonstrates bursts of muscle activity in *both* phases (see Figures of *Forman et al., 2015* and Figure 1B of *Lockyer et al., 2018*). From a motor control standpoint, this
may indicate that greater co-contraction is needed during the flexion phase to oppose the biceps brachii/stabilize the elbow joint. This also suggests that the elbow flexors and extensors are driven by unique motor control strategies. However, these statements are limited in that muscle activity from other upper-limb muscles have not been characterized between the two phases of arm cycling. Additionally, the unique phase characteristics of the biceps and triceps brachii were observed in studies utilizing just 25 W (*Forman, Monks & Power, 2019*; *Forman et al., 2015*). Not only are these workloads low, they are also absolute power outputs, which likely induce a greater variation of muscle activity between individuals than relative workloads. It is presently unclear if these phase differences persist, or are perhaps modulated, at higher relative arm cycling intensities.

There are only three studies that have shown the influence of workload on EMG of the arm musculature during cycling (*Bernasconi et al., 2006*; *Hundza et al., 2012*; *Spence et al., 2016*). While EMG increased in each study with increased workload, as would be expected, specific information such as flexion/extension phase-dependence or activation pattern was not provided (*Bernasconi et al., 2006*; *Hundza et al., 2012*) and/or there were minimal workloads utilized (*Spence et al., 2016*). The objective of the present study was to characterize arm muscle activity (i.e., iEMG amplitude) at different relative workloads during two different phases of arm cycling, flexion and extension, as defined by movement at the elbow. We hypothesized that constant cadence cycling at different mechanical loads (i.e., different power outputs) would result in: (1) increased iEMG amplitude (2) phase-dependent differences in iEMG amplitudes (3) a linear relationship between iEMG amplitude and power output and (4) differences in iEMG amplitudes between an absolute 25 W workload and a low relative workload of just 5% of peak power output (PPO).

## METHODOLOGY

### Ethical approval

The procedures of the experiment were verbally explained to each volunteer prior to the start of the session. Once all questions were answered, written consent was obtained. This study was conducted in accordance with the Helsinki declaration and approved by the Interdisciplinary Committee on Ethics in Human Research at Memorial University of Newfoundland (ICEHR#: 20150140-HK). Procedures were in accordance with the Tri-Council guidelines in Canada and potential risks were fully disclosed to participants.

### Participants

Eleven healthy individuals (six males and five females, 25.2 ± 4.4 years of age, 73.6 ± 7.8 kg, nine right-hand dominant, two left-hand dominant) were recruited for this study. Participants had no known neurological impairments. Prior to the experiment, all participants completed a Physical Activity Readiness Questionnaire (PAR-Q+) to screen for any contraindications to exercise or physical activity and an Edinburgh Handedness Inventory checklist to quantify hand dominance. Participants were required to refrain from any heavy exercise, especially upper body exercise, 24 h prior to the start of testing.
## Experimental procedure

Participants attended a familiarization session to practice arm cycling sprints that were required during the experimental session to determine peak power output (PPO). This session was followed by an experimental session with a minimum of 24 h between. During the experimental session participants first completed a 5-minute warm-up using a Monark cycle ergometer (Ergomedic 894 E), with only the 1 kg weighted basket as resistance, at a self-selected pace. The ergometer was securely mounted to the top of a table and fitted with hand pedals. Following the warm-up and a 5-min rest break, participants performed a 10 s maximal arm ergometry sprint using 5% of their body weight as the resistance to determine PPO. The Monark cycle ergometer recorded power output at a sampling rate of 50 Hz; the highest power output of a single sample throughout the 10 s trial was deemed to be the individual's PPO. Results of this cycling trial were then used to determine the relative intensity for all subsequent trials. Following a minimum 10 min post-sprint rest period, participants were moved to a SCIFIT cycle ergometer (model PRO2 Total Body) to perform arm cycling at 11 different intensities, 10 of which were made relative to the PPO and one which was done at 25 W. The 25 W condition was constant for all participants, given that 25 W is a common workload used during arm cycling studies (*Bressel et al., 2001*; *Forman et al., 2014*). The remaining 10 trials were randomized and performed at relative intensities ranging from 5–50% of the PPO. For all trials participants cycled at a constant cadence of 60 rpm for 20 s.

## Experimental set-up

Participants were seated upright at a comfortable distance from the hand pedals, so that during cycling, there was no reaching or variation in trunk posture (Fig. 1). To further ensure that posture was maintained throughout all trials, each participant was strapped securely to the ergometer seat with straps placed over the shoulders and across the chest. Movement of the shoulders and arms was not impeded. The hand pedals of the ergometer were fixed 180 degrees out of phase and the seat height was adjusted so that the shoulders of each individual were approximately the same height as the centre of arm crank shaft. Participants gripped the ergometer handles with the forearms in a pronated position.

Cycle crank positions were made relative to a clock face (12, 3, 6, and 9 o'clock, as viewed from the right crank arm) with the "top dead centre" position of the crank arm defined as 12 o'clock and "bottom dead centre" as 6 o'clock, which is identical to previous investigations (*Carroll et al., 2006*; *Klimstra, Thomas & Zehr, 2011*; *Nippard et al., 2020*; *Power et al., 2018*; *Zehr et al., 2004*; *Zehr & Chua, 2000*). The biceps brachii and triceps brachii were the main muscles of interest, thus the terminology used to describe the cycling movement is based on the position of the dominant elbow joint. Elbow flexion was defined as the movement from 3 to 9 o'clock, while the hand was moving toward the body. Elbow extension was defined as the movement from 9 to 3 o'clock, while the hand was moving away from the body. There were magnets positioned at 3 o'clock and 9 o'clock on the SciFit Bicycle in order to enable crank position to be tracked during cycling. When the crank passed the magnets at the 3 o'clock and the 9 o'clock positions, a 5 volt pulse was sent from

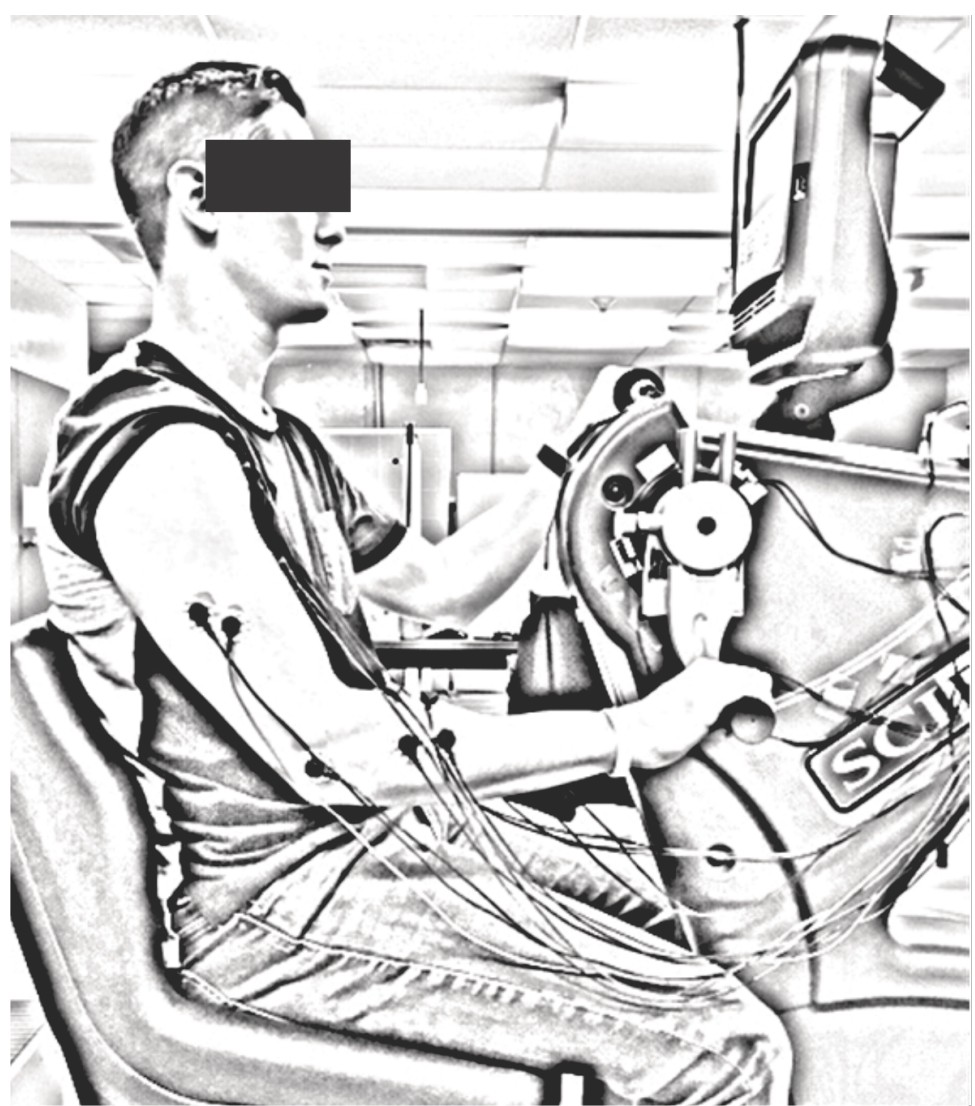

**Figure 1** **Example of the experimental setup.** Participants were seated with their shoulders at approximately the same height as the axis of the crank shaft on the SCIFIT cycle ergometer while cycling at a constant cadence of 60 rpm at 11 different workloads (5–50% of PPO and 25 W). Flexion and extension were made relative to the elbow joint (flexion from 3 to 9 o'clock and extension from 9 to 3 o'clock; both relative to a clock face; arm in the example at the 6 o'clock position). iEMG was recorded from biceps brachii, triceps brachii, anterior deltoid, brachioradialis, FCR and ECR from the dominant limb.

the SciFit Bicycle to the data collection software. This pulse was recorded and used to track crank position through-out all cycling trials.

## Electromyography recording

EMG of the biceps brachii, lateral head of the triceps brachii, anterior deltoid, brachioradialis, flexor carpi radialis (FCR), and extensor carpi radialis (ECR) of the dominant arm were recorded using pairs of surface electrodes (Medi-Trace 130 ECG conductive adhesive electrodes). The inter-electrode distance was two cm and all electrodes

were aligned to fiber direction of the target muscles. A ground electrode was placed on the lateral epicondyle. Prior to electrode placement the skin was thoroughly prepared by shaving any hair and the removal of dead epithelial cells (using abrasive paper) followed by sanitization with an isopropyl alcohol swab. Muscle activation data was collected at 2,000 Hz using the BIOPAC MP-100 data acquisition system with Acknowledge 4 software and an EMG100C differential amplifier (CMRR 110dB (50/60 Hz), input impedance 2 MΩ, bandpass filter 10 Hz–500 Hz). Data obtained during the experiment were analyzed offline using code written in Visual Basic.

## Data analysis

The EMG data were amplitude normalized by dividing the raw EMG during cycling by the muscle specific maximum EMG from the 10-second maximal arm ergometry sprint. The maximum EMG amplitude was determined using a 100 ms RMS moving window (as per *Burden & Bartlett, 1999*) to process the raw EMG from each muscle over the duration of the 10-second sprint. The resulting smoothed signal was examined to determine the peak EMG for each muscle, which was then used to normalize the amplitude of all sub-maximal cycling trials.

The submaximal cycling trials were then analyzed by examining the middle 10 seconds of data from each trial. These 10 seconds of data were divided in to sections that represented one complete revolution of the crank handle (from 3 o'clock to 3 o'clock). Each revolution was further broken down in to an elbow flexion phase (from 3 o'clock to 9 o'clock) and an elbow extension phase (from 9 o'clock to 3 o'clock). This was done using the magnet signal described above. For most individuals, a total of 10 revolutions were completed during the 10 seconds of cycling. Once the data was windowed, integrated EMG (iEMG) was calculated for the following time periods: the full revolution, the flexion phase and the extension phase. Trapezoid rule was used for these calculations.

To assist with the visual presentation of the data, linear envelope, ensemble average EMG was calculated for each arm cycling intensity. This was done using the following steps:

- Raw, amplitude normalized and windowed EMG was full wave rectified and low pass filtered at 10 Hz using a fourth order dual-pass butterworth filter. Data from each of the complete revolutions were then rubberbanded to normalize it to time. One revolution was considered 100% of the whole cycle with the time period from 3–9 being fit to the first 50% (flexion) of the rubberbanded signal and 9–3 to the last 50% (extension).
- These rubberbanded trials were then ensemble averaged across all trials for each intensity. The end result was an average linear envelope for each muscle at each intensity.

## Statistics

All statistical analysis was performed using IBM's SPSS Statistics Version 23. Assumptions of sphericity were tested using the Mauchley test, and if violated, the Greenhouse-Geisser estimates of sphericity correction was applied to the degrees of freedom. The data were normally distributed as determined using the Kolomogorov-Smirnov normality test. Separate two-way (PHASE x INTENSITY) repeated-measures ANOVAs were used to

assess the iEMG of each muscle ($n = 11$ for biceps and triceps brachii and $n = 5$ for the remaining muscles) during two phases (flexion and extension) and 11 different workloads (25 W and percentages of PPO). To determine whether the relationship between iEMG and intensity was best described as linear during both phases of arm cycling, a series of twelve repeated-measures one-way ANOVAs were conducted for each muscle examined using Polynomial Contrasts (i.e., linear, quadratic or cubic). Trends were determined by examining the $F$-values of each of the 3 models as well as the observed power. All statistics were run on group data and a significance level of $p < .05$ was used. All data are reported in text as means $\pm$ SD and illustrated as means $\pm$ SE in figures for clarity.

## RESULTS

### EMG activity patterns of arm muscles during arm cycling

The EMG activity (i.e., linear envelope ensemble averaged EMG) for each of the arm muscles recorded during arm cycling are plotted in Fig. 2. In this figure we have omitted EMG recorded from 5 of the 11 intensities (i.e., 5, 15, 25, 35 and 45% of PPO) for figure clarity. There are several distinct qualitative features worth noting. First, the elbow flexor muscles, biceps brachii and brachioradialis are very active during elbow flexion (3 to 9 o'clock), while they are virtually silent during elbow extension (9 to 3 o'clock; Figs. 2A and 2B). The triceps brachii appears to be highly active during the extension phase, however, as opposed to the biceps brachii there is more of a biphasic activation pattern, with the muscle also being active during the flexion phase (Fig. 2C). FCR also appears to be biphasically active while ECR appears to have a peak activation occurring at roughly the 9 o'clock position, corresponding to the end of flexion/beginning of extension (Figs. 2D and 2E, respectively). The anterior deltoid is more active during the elbow extension phase (Fig. 2F), though it begins to activate during the latter portion of elbow flexion (i.e., approximately 6 o'clock). It is also clear that as the intensity of cycling increased so too did the EMG activation level in each muscle.

### Intensity- and phase-dependent effects on iEMG of recorded muscles during arm cycling

Table 1 summarizes the findings for each of the muscles recorded. All muscles demonstrated a significant main effect of INTENSITY on iEMG. With the exception of the triceps brachii and ECR, muscles also demonstrated significant main effects for PHASE (i.e., elbow flexion vs extension) and an interaction effect between INTENSITY and PHASE.

#### *Biceps brachii*

A significant INTERACTION effect was shown with iEMG at each cycling intensity being significantly different between flexion and extension; flexion had higher iEMG at all levels of intensity compared to extension. During *flexion*, iEMG at 25 W was not statistically different than 5% PPO, though there was a trend for higher activation during the 5% PPO cycling ($p = 0.052$). The iEMG at subsequent ascending workloads significantly increased from 5 to 35% PPO. Non-significant increases in iEMG occurred between 35 to 50% PPO (i.e., iEMG recorded at 35, 40, 45 and 50% PPO were not significantly different from each

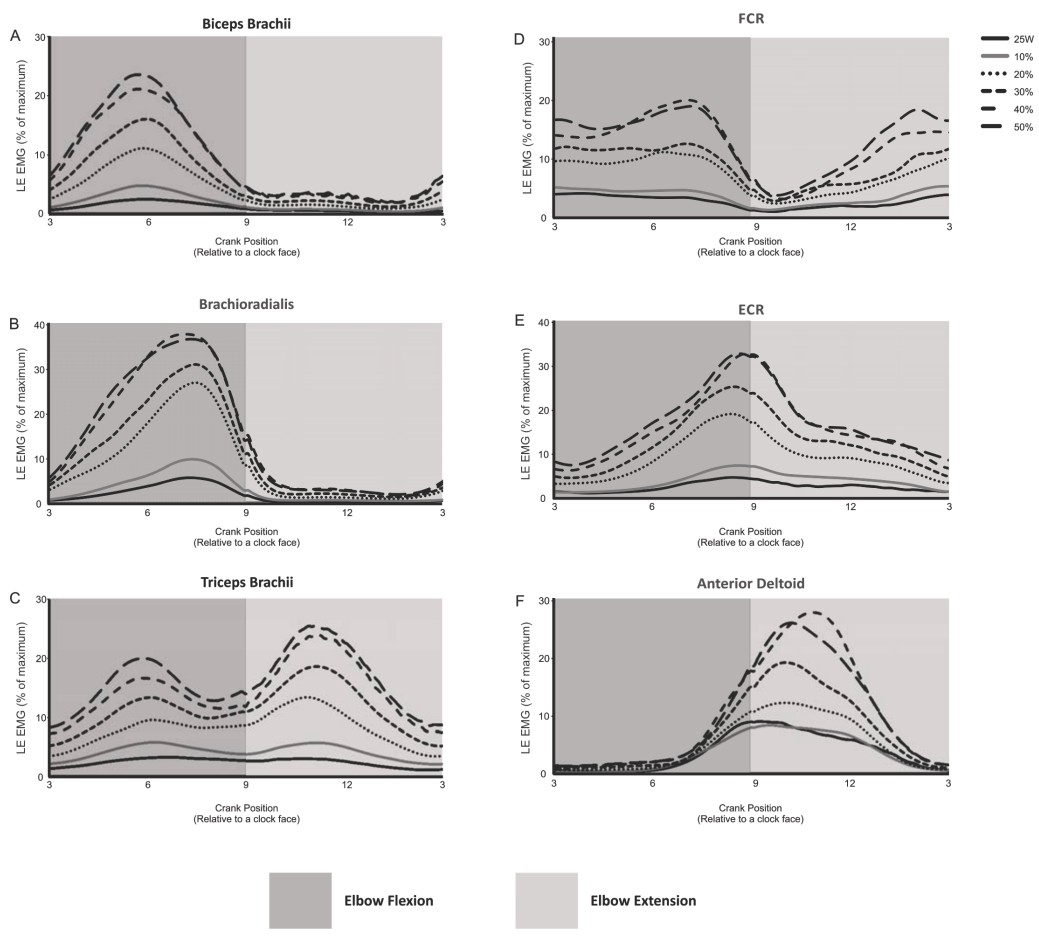

**Figure 2 Linear envelope ensemble averaged iEMG during 1 full revolution for each muscle examined.** Arm cycling intensities shown include the absolute power output (25 W) and the percentages of maximum PPO (10, 20, 30, 40 and 50%). The remaining intensities of arm cycling are excluded for figure clarity (5, 15, 25, 35 and 45% PPO). Amplitudes are expressed as a percentage of maximal EMG. The dark gray region in each section indicates elbow flexion (3 to 9 o'clock) while the light gray region represents elbow extension (9 to 3 o'clock).

**Table 1 iEMG and workload.** Statistical summary table.

| Muscle | Position Main Effect | Intensity Main Effect | Interaction Main Effect |
|---|---|---|---|
| Biceps Brachii | $(F_{(1,10)} = 105.363, p < .001)$ | $(F_{(2.72, 27.22)} = 59.435, p < .001)$ | $(F_{(2.98, 29.18)} = 41.737, p < .001)$ |
| Triceps Brachii | $(F_{(1,10)} = 1.362, p = .270)$ | $(F_{(2.06, 20.6)} = 65.015, < .001)$ | $(F_{(1.51, 15.16)} = 2.246, p = .148)$ |
| Anterior Deltoid | $(F_{(1,4)} = 17.067, p = .014)$ | $(F_{(10,40)} = 15.110, p = .001)$ | $(F_{(10,40)} = 15.039, p = .001)$ |
| Brachioradialis | $(F_{(1,4)} = 37.097, p = .004)$ | $(F_{(10,40)} = 37.954, p < .001)$ | $(F_{(10,40)} = 28.886, p < .001)$ |
| FCR | $(F_{(1,4)} = 10.484, p = .032)$ | $(F_{(10,40)} = 56.171, p < .001)$ | $(F_{(10,40)} = 4.772, p = .040)$ |
| ECR | $(F_{(1,4)} = .217, p = .665)$ | $(F_{(10,40)} = 27.585, p < .001)$ | $(F_{(10,40)} = .509, p = .579)$ |

other though they were greater than all intensities $\leq$ 30% PPO). During *extension*, changes in iEMG were similar to that during flexion, with iEMG at 25 W and 5% PPO being not statistically different and subsequent workloads in ascending order significantly increased

from 5 to 35% PPO. Non-significant increases in iEMG occurred between 35 to 50% PPO (i.e., iEMG recorded at 35, 40, 45 and 50% PPO were not significantly different from each other though they were greater than all intensities ≤ 30% PPO).

### Brachioradialis

A significant INTERACTION effect was shown with iEMG at each cycling intensity being significantly different between flexion and extension; flexion had higher iEMG at all levels of intensity compared to extension. During *flexion*, iEMG in the brachioradialis could be grouped into three 'blocks' (block 1, 25 W to 10% PPO; block 2, 20 to 30% PPO; block 3, 40 to 50% PPO) and two stand-alone intensities (15 and 35% PPO). The iEMG recorded within each block were not different from each other but were different from the other blocks and also the stand-alone intensities. During *extension*, there were no differences between 25 W, 5 and 10% PPO. The iEMG during subsequent workloads significantly increased from 15 to 35% PPO. Similar to the biceps brachii, non-significant increases in iEMG occurred between 35 to 50% PPO (i.e., iEMG recorded at 35, 40, 45 and 50% PPO were not significantly different from each other though they were greater than all intensities ≤ 30% PPO).

### Triceps brachii

There was a significant main effect for INTENSITY. With the data collapsed for PHASE, a similar finding to that of the biceps brachii was revealed. Specifically, iEMG at 25 W was not statistically different than 5% PPO but iEMG in subsequent workloads showed significant increases from 5 to 35% PPO (i.e., iEMG recorded at 35, 40, 45 and 50% PPO were not significantly different from each other though they were greater than all intensities ≤ 30% PPO).

### FCR

A significant INTERACTION effect was shown for iEMG, with each cycling intensity being significantly different between flexion and extension, with the exception of the 15 and 50% PPO ($p = .074$ and $p = .124$, respectively) intensities; flexion had higher iEMG at all other levels of intensity compared to extension. Within both *flexion and extension*, iEMG at 25 W, 5 and 10% PPO were not statistically different from one another. Subsequent workloads in ascending order did not significantly increase from 10 to 50% PPO.

### ECR

There was a significant main effect for INTENSITY (see Table 1) but not PHASE, thus data was collapsed to examine the general influence of cycling INTENSITY on iEMG. There were no significant differences in iEMG between 25 W, 5 and 10% PPO. Beginning at 15% PPO iEMG significantly increased with subsequent increases in PPO up to and including 30% PPO. iEMG from 30 to 50% PPO were not statistically different from each other.

### Anterior deltoid

During *flexion,* iEMG at 25 W, 5 and 10% PPO were not statistically different from one another. Subsequent workloads in ascending order significantly increased from 10 to 25% PPO. Non-significant increases in iEMG occurred from 25 to 50% PPO (i.e., iEMG

recorded at 25, 30, 35, 40, 45 and 50% PPO were not significantly different from each other though they were greater than all intensities ≤ 20% PPO). During *extension*, iEMG at 25 W, 5 and 10% PPO were not statistically different from one another. Subsequent workloads in ascending order significantly increased between 10 and 20% PPO (i.e., 20>15>10). Non-significant increases in iEMG occurred from 20 to 50% PPO (i.e., iEMG recorded at 20, 25, 30, 35, 40, 45 and 50% PPO were not significantly different from each other though they were greater than all intensities ≤ 20% PPO). As for the INTERACTION effect, iEMG at each cycling intensity was significantly different between flexion and extension, with the exception of the 5% PPO ($p = .058$); extension had higher iEMG at all levels of intensity compared to flexion.

### The EMG-power output relationship during arm cycling is linear

Figures 3A–3F shows that the relationship between iEMG and workload was linear for all muscles examined during both the flexion and extension phases of arm cycling, with details provided in Table 2. We also compared the slopes of the linear relationships between flexion and extension for each muscle using a paired $t$-test to assess if the gain in iEMG was different between phases within a muscle. The slope was significantly different between flexion and extension for each muscle examined ($p < 0.001$ for biceps brachii, brachioradialis, FCR and anteriod deltoid; $p = 0.003$ for triceps brachii) with the exception of ECR ($p = 0.07$).

## DISCUSSION

This is the first study to characterize the flexion/extension specificity of arm muscle activity during arm cycling over a wide range of power outputs. To do so we divided the arm cycling motor output into two phases, flexion and extension, made relative to the elbow joint. Using this criteria the most important results in the present study are: (1) arm muscle activity during arm cycling at different relative PPOs was quantified, (2) iEMG amplitude increased significantly with increased power outputs, (3) iEMG differed between the flexion and extension phases in all muscles except the triceps brachii and ECR, and (4) a linear iEMG-power output relationship for each of the muscles examined.

### iEMG amplitudes during arm cycling

The basic pattern of arm muscle activity during arm cycling has been previously described (*Bressel et al., 2001*; *Zehr & Chua, 2000*) but few studies have examined the influence of workload on EMG of the arm muscles during arm cycling (*Bernasconi et al., 2006*; *Hundza et al., 2012*; *Smith et al., 2008*; *Spence et al., 2016*) . *Bernasconi et al. (2006)* reported an increase in EMG with increased workload in each of the muscles examined (biceps brachii, triceps brachii, anterior deltoid, and infraspinatus) during an arm cycling VO $_2$ max test. Their objective, however, was not to give a detailed description of muscle activation levels as arm cycling intensity increased. For example, they did not examine EMG during the two propulsive phases of arm cycling, nor did they show representative traces of EMG activity. Similar findings were reported by *Hundza et al. (2012)*, (i.e., increased EMG with increased arm cycling workloads). As was the case in the Bernasconi report, however, they

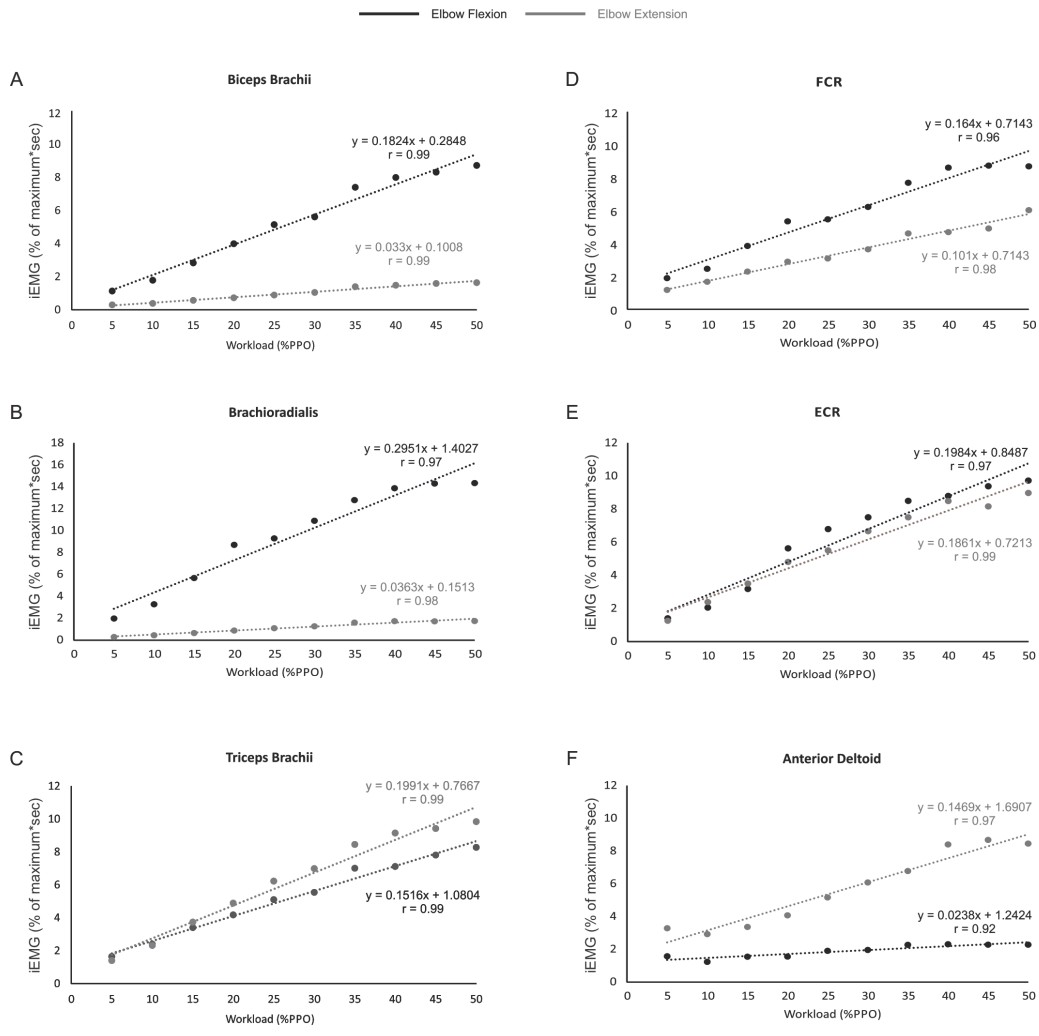

**Figure 3** **The relationship between iEMG and power output for each muscle during the elbow flexion and extension phases of arm cycling.** The slopes and r values are illustrated for each position in each muscle. Slopes were significantly different for muscles shown in (A–D) and (F). While the slopes were not significantly different between flexion and extension in ECR ($p = 0.07$; Fig. 3F), they are plotted to allow visual inspection.

did not assess the phase-dependence or pattern of arm muscle EMG, as that was not their intent. *Smith et al. (2008)* also demonstrated that muscle activity increased as power output increased from 50 to 100 W. However, asynchronous versus synchronous cycling was the primary focus of this study, and with only two absolute workloads, complete details into a power and muscle activity relationship were not elucidated. Finally, a recent study from our lab examined corticospinal excitability to the biceps and triceps brachii during arm cycling at two different workloads (*Spence et al., 2016*). In that study, however, only the biceps and triceps brachii EMG were reported at two different workloads (5 and 15% PPO) with EMG recordings made at mid-elbow flexion and extension as previously discussed.

**Table 2  Relationships between iEMG and workload.** Statistical summary table.

| Muscle | Phase | Linear | Quadratic | Cubic |
|---|---|---|---|---|
| Biceps Brachii | F | $102.766_{(1,10)}, p < .001$ | $3.138_{(1,10)}, p = .107$ | $4.669_{(1,10)}, p = .056$ |
| Biceps Brachii | E | $83.347_{(1,10)}, p < .001$ | $.541_{(1,10)}, p = .479$ | $10.114_{(1,10)}, p = .010$[a] |
| Triceps Brachii | F | $41.255_{(1,10)}, p < .001$ | $2.047_{(1,10)}, p = .183$ | $1.257_{(1,10)}, p = .288$ |
| Triceps Brachii | E | $85.090_{(1,10)}, p < .001$ | $3.653_{(1,10)}, p = .085$ | $3.764_{(1,10)}, p = .081$ |
| Anterior Deltoid | F | $20.378_{(1,4)}, p < .05$ | $.172_{(1,4)}, p = .700$ | $3.681_{(1,4)}, p = .127$ |
| Anterior Deltoid | E | $28.356_{(1,4)}, p < .01$ | $.222_{(1,4)}, p = .662$ | $12.342_{(1,4)}, p = .025$ |
| Brachioradialis | F | $85.296_{(1,4)}, p = .001$ | $7.819_{(1,4)}, p = .049$ | $1.760_{(1,4)}, p = .255$ |
| Brachioradialis | E | $40.878_{(1,4)}, p < .01$ | $6.775_{(1,4)}, p = .060$ | $23.252_{(1,4)}, p = .009$ |
| FCR | F | $449.438_{(1,4)}, p < .001$ | $5.663_{(1,4)}, p = .076$ | $1.584_{(1,4)}, p = .277$ |
| FCR | E | $25.641_{(1,4)}, p < .01$ | $.000_{(1,4)}, p = .995$ | $.573_{(1,4)}, p = .491$ |
| ECR | F | $49.141_{(1,4)}, p < .01$ | $2.233_{(1,4)}, p = .209$ | $1.236_{(1,4)}, p = .329$ |
| ECR | E | $19.137_{(1,4)}, p < .05$ | $2.342_{(1,4)}, p = .201$ | $1.654_{(1,4)}, p = .268$ |

Notes.
[a] $n = 11$ for biceps and triceps brachii and $n = 5$ for the remaining muscles. F, flexion phase and E, extension phase.

The present work, however, is the first to show patterns of arm muscle activity over a wide range of power outputs during arm cycling (Fig. 2). Similar to previous work (*Zehr & Chua, 2000*) reciprocal activation of functional antagonists was observed between muscles acting at the wrist (FCR and ECR; Figs. 2D and 2E, respectively) and to a certain degree at the elbow (i.e., during elbow extension but not flexion; see biceps and triceps brachii traces in Figs. 2A and 2C). Given our laboratory's interest in the neural control of the biceps and triceps brachii musculature during arm cycling (*Copithorne, Forman & Power, 2015*; *Forman et al., 2014*; *Forman et al., 2015*; *Forman et al., 2016a*; *Forman et al., 2016b*; *Power & Copithorne, 2013*; *Spence et al., 2016*), we were particularly interested in the phase- and workload-dependent changes in those muscles. Interestingly, we show that the elbow flexors (i.e., biceps brachii and brachioradialis) demonstrated strong phase-dependence in EMG whereas the triceps brachii did not (compare Figs. 2A and 2B with 2C). This finding is partially a function of the how we defined flexion and extension and thus during which portions of the arm cycling revolution measurements were made. In our previous work, for example, we assessed EMG amplitude at mid-elbow flexion (6 o'clock) and mid-elbow extension (12 o'clock) (*Forman et al., 2014*; *Forman et al., 2016a*; *Forman et al., 2016b*). These points in time during a full cycle represent very different EMG activity levels than those averaged over half a revolution as was done in the present study (see Figs. 2A and 2C) and closely align with peak activation in the biceps (~5–6 o'clock) and triceps brachii (~10–11 o'clock). In the present study we separated and assessed iEMG activity during two phases, flexion (3 to 9 o'clock) and extension (9 to 3 o'clock) relative to the elbow joint and not at specific points of the cycle.

Why do the biceps and triceps brachii demonstrate different phase-dependence characteristics? The biceps brachii is bi-articular (i.e., crossing two joints—elbow and shoulder) and contributes to elbow flexion. Thus, as expected, the biceps brachii was highly active during elbow flexion and relatively inactive during elbow extension. The long head of the triceps brachii is bi-articular (i.e., originates from the infraglenoid tubercle of

the scapular, lateral and medial head from the humerus), while the medial and lateral heads of the triceps brachii are mono-articular activing only at the elbow. Because we recorded from the lateral head of the triceps brachii it was surprising to find that its activity level was not different between the elbow flexion and extension phases of arm cycling given its role in elbow extension. Though not assessed in the same manner (i.e., elbow flexion vs extension), the triceps brachii muscle appears to demonstrate a stronger phase-dependence in the study by *Zehr & Chua (2000)*, though the head of the triceps from which the recordings were made is not stated in that manuscript (*Zehr & Chua, 2000*). There are several potential explanations for our present finding. First, arm cycling was performed with the hand in a pronated position. Though forces at the hand pedal were not assessed, it is likely that during arm cycling the elbow extensors are active during elbow flexion in an attempt to push the hand down on the pedal to maintain a constant grip. Thus, the lateral head of the triceps may act as an extensor during elbow extension, while during elbow flexion, it may act to stabilize the hand. Second, during elbow flexion the long head of the triceps acts as a prime mover to extend the shoulder (in addition to the rear deltoid and latissimus dorsi). Because its primary role in the flexion phase of arm cycling is likely shoulder extension, its capacity to stabilize the elbow may be limited. Therefore, it is possible that the lateral and medial heads of the triceps brachii assume the role of elbow stabilization, resulting in elevated iEMG in the flexion phase. A switch to a larger role during elbow extension for the lateral head of the triceps may occur as the workload increases as suggested via the higher gain in the iEMG-power output relationship, specifically at higher workloads (see Fig. 3C). Finally, activation of the lateral head of the triceps brachii may be partially due to task novelty, resulting in unnecessary co-contraction. Recent work showed a similar bi-phasic activation pattern of the triceps brachii that was abolished following arm cycling training in persons with spinal cord injury (i.e., the triceps brachii activity was absent during flexion) (*Brousseau et al., 2016*), suggesting that a learning response may occur.

### iEMG/power-output is linear for arm muscles during arm cycling

As expected, muscle activation levels (i.e., iEMG amplitudes) generally increased as arm cycling workload increased for all muscles examined (see Table 1 and Fig. 3). The increases in iEMG amplitudes reflect the increased recruitment and firing frequency of the motor units from which we recorded. Based on the size principle (*Adrian & Bronk, 1929*; *Henneman, 1957*) as the cycling intensity increased, additional motor units, including larger, faster motor units, would be recruited to assist with force production resulting in an increase in the iEMG amplitude. While the iEMG-power output relationship was linear for each muscle, there are some points that should be mentioned. It was noted that iEMG amplitudes were not significantly different when comparisons are made *within* the lowest (generally 25 W vs 5% PPO) and highest (generally 35–50% PPO) cycling intensities for several muscles (see Results). While the lack of difference at the lower intensities may simply be due to the fact that the cycling intensities were not high enough, it is interesting that there was a general plateau of the iEMG at the higher power outputs given that the highest workload was set at only 50% of PPO. This may be due to confounding variables related to how the iEMG was normalized (see *Methodological Considerations*)

or the result of torso or lower extremity muscles making larger contributions to motor output execution at higher relative PPO (*Smith et al., 2008*). Physiologically the plateau may be related to motoneurone pools being near full recruitment and/or maximal firing rate. During isometric contractions, for example, spinal motoneurone excitability as assessed via responsiveness to transmastoid electrical stimulation plateaus at ~50% of maximal voluntary effort (*Pearcey, Power & Button, 2014*). The continued increase in power output likely involves changes in muscle co-ordination and/or muscle synergies (*Blake & Wakeling, 2015*; *Wakeling, Blake & Chan, 2010*), keeping in mind that arm cycling is a bilateral motor output that involves multiple muscles, many of which were not examined in the present study. These findings are in general agreement with previous work using isometric contractions to characterize the EMG force relationship. Studies assessing the relationship between force and EMG have shown that as workload increases, EMG also increases both linearly (*Lippold, 1952*; *Woods & Bigland-Ritchie, 1983*) and non-linearly (*Woods & Bigland-Ritchie, 1983*). Importantly, *Moritani & deVries (1978)* reported a linear relationship between workload and EMG in the right elbow flexor muscles during several submaximal contractions.

## Gain of the iEMG-power output relationship during elbow flexion and extension phases

The slopes of the linear relationships between iEMG amplitudes and power output were different between elbow flexion and extension for each muscle with the exception of ECR ($p = 0.07$). The most interesting finding here is that the triceps brachii 'gain' was higher during elbow extension than flexion (i.e., as the power output increased there was a greater increase in iEMG amplitude during elbow extension than flexion). This may relate to our thought that the triceps brachii is active during flexion at lower intensities as a hand stabilizer and that as the intensity of arm cycling increases the triceps brachii is recruited to produce elbow extension forces to a greater degree to assist with arm crank movement.

## Methodological considerations

There are several methodological considerations to be taken considered when interpreting the present results. The data collected in this study occurred over a relatively brief period (only 20 s). As the duration of sustained exercise increases, so too does the amplitude of EMG signals (*Takaishi et al., 1996*). Had the trials been longer in the present study, the observed muscle activity patterns may have been different. It is also important to note the difference in cycling cadence between the maximal ergometry sprint and the cycling trials using different percentages of PPO. During the maximal sprint, participants cycled as fast as they could against a set resistance whereas a 60 rpm cadence was used during all submaximal trials in which EMG was assessed. This likely lead to a much higher level of muscle activity during the cycling sprint than could be obtained while cycling at a slower cadence due to the added influence of cadence-dependent changes in descending drive and/or afferent feedback. This is an important consideration because the cycling trials are normalized to the maximal sprint EMG, which partially explains the relatively low level of EMG recorded from the muscles during the relative workloads, even at 50% PPO. It is also important to

recognize that arm cycling is a complex motor output that involves the activation of many muscles working at several joints, bilaterally; at higher PPO intensities, contributions from the torso and lower limbs are likely to be significant as well (*Smith et al., 2008*). This means that the timing and co-ordination of muscle activity and/or muscle synergies are likely of great importance when considering how the neuromuscular system produces this motor output, particularly as power output increases (*Blake & Wakeling, 2015*; *Enders, V & Nigg, 2015*; *Wakeling, Blake & Chan, 2010*). We did not assess timing or synergies in the present study as our goal was to characterize flexion/extension phase-dependent EMG amplitudes as power output increased given our interest in how cycling intensity alters corticospinal excitability, which is dependent on the amount of EMG activity produced. Also, the present study only examined muscle activity in two distinct phases of arm cycling–flexion and extension relative to elbow joint movement. While this was ultimately central to one of our main objectives, arm cycling can be characterized by more than just two phases. Indeed, as can be seen in Fig. 2, the muscle activity patterns of some muscles change constantly throughout a full revolution. Had data been broken down into a greater number of phases, a more complex phase-dependent behaviour would likely have been observed for all examined muscles.

Lastly, two different ergometers (Monark and SCIFIT) were utilized in this study to fulfill two separate needs (establish PPO and set stable target power outputs). These two needs could not be addressed by one device alone. It is therefore possible that the power output measurement precision between these devices was slightly different. If so, the relative targets used in this study (5–50% of PPO) may not have been precisely as stated. However, such differences were almost certainly small and were unlikely to have influenced the main outcomes of this study.

## CONCLUSION

The present study characterized the iEMG pattern of activation during arm cycling at different relative intensities. One of the main findings was a linear relationship between iEMG amplitude and power output during arm cycling. We also showed that the influence of the flexion/extension cycling phases were muscle-dependent (i.e., muscle activity differed between these two phases in some muscles but not in others). Given the well-known impact of EMG amplitude on various measures of neural excitability it may be of importance for individuals to cycle at relative intensities (i.e., percentages of their maximal PPO).

## ACKNOWLEDGEMENTS

The authors thank Michael Monks for assistance with data collection, Dr. Tim Alkanani for technical support and all of the volunteer participants.

### Funding
This work was supported by a Discovery Grant to Kevin Power from the Natural Sciences and Engineering Research Council of Canada (NSERC: #RGPIN-2016-03646). The funders had no role in study design, data collection and analysis, decision to publish, or preparation of the manuscript.

### Grant Disclosures
The following grant information was disclosed by the authors:
Discovery Grant to Kevin Power from the Natural Sciences and Engineering Research Council of Canada:  #RGPIN-2016-03646.

### Competing Interests
The authors declare there are no competing interests.

### Author Contributions
- Carla P. Chaytor conceived and designed the experiments, performed the experiments, analyzed the data, prepared figures and/or tables, authored or reviewed drafts of the paper, and approved the final draft.
- Davis Forman conceived and designed the experiments, performed the experiments, authored or reviewed drafts of the paper, and approved the final draft.
- Jeannette Byrne and Angela Loucks-Atkinson conceived and designed the experiments, analyzed the data, authored or reviewed drafts of the paper, and approved the final draft.
- Kevin E. Power conceived and designed the experiments, analyzed the data, prepared figures and/or tables, authored or reviewed drafts of the paper, and approved the final draft.

### Human Ethics
The following information was supplied relating to ethical approvals (i.e., approving body and any reference numbers):

This study was conducted in accordance with the Helsinki declaration and approved by the Interdisciplinary Committee on Ethics in Human Research at Memorial University of Newfoundland (ICEHR#: 20150140-HK). Procedures were in accordance with the Tri-Council guideline in Canada and potential risks were fully disclosed to participants.

### Data Availability
Data are available as a Supplementary File.

### Supplemental Information
Supplemental information for this article can be found online at http://dx.doi.org/10.7717/peerj.9759#supplemental-information.

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
