# Peer review of "Changes in muscle activity during the flexion and extension phases of arm cycling as an effect of power output are muscle-specific"

_PeerJ, doi:10.7717/peerj.9759_

## Round 0.1 · original submission · Minor Revisions

· Academic Editor

Minor Revisions

I received two reviews of your study. Both reviewers agree that your ms conveys sound research. I would like you to clarify whether your data were acquired from all participants for all muscles. Reviewer 1 has concerns because you consider only two phases to interpret that TB and ECR iEMG are not modulated by phase. This seems an overstatement. Reviewer 2 asks you to explain your rationale for using a participant’s performance during an all-out sprint test to prescribe exercise intensities. Both reviewers requested more details about your methodology. Please take these and all other suggestions in full consideration before resending your ms.

·

Basic reporting

The manuscript is well-written with professional English grammar throughout. The literature cited is relevant and provides the relevant background information required to introduce the topic. The results are well-aligned with the hypothesis of the authors. I have only one comment about a reference choice below.

Line 53 - Klarner et al 2016: This study is about arm and leg cycling - perhaps the authors meant to reference Kaupp et al 2018 from the same group?

Additionally, the figure resolutions seem to be very low. Perhaps a higher resolution could be used?

Experimental design

The concept of a linear relationship between EMG amplitude of many muscles and cycling intensity is novel and well-explored in this manuscript. One main point requires clarification, however.

It is unclear whether data was acquired from all participants for all muscles. In the supplemental table, it only has data from 6 participants for muscles other than TB and BB. Can the authors please clarify?

Validity of the findings

The conclusions are fine, with the exception of one point, which arises in three places: 1) abstract, 2) beginning of the discussion and 3) in the conclusion paragraph. The authors conclude that TB and ECR iEMG are not modulated by phase and this might be overstated. Perhaps it would be more suited to state that iEMG did not differ between elbow flexion and extension phases of cycling, rather than that it is not phase-dependent. Currently, the authors acknowledge that they only measured two phases by saying “In the present study we separated and assessed iEMG activity during two phases, flexion (3 to 9 o’clock) and extension (9 to 3 o’clock) relative to the elbow joint and not at specific points of the cycle” but this is still problematic for the determination of whether a muscle exhibits phase-dependent modulation of muscle activity. As the number of bins decreases, the resolution of phasic modulation of muscle activity is lost. Many authors in the past have analyzed rhythmic motor output with varying numbers of bins (or phases) within the movement cycle that range from 6 (Marigold et al 2017 Exp Br Res) to 16 bins (Hoogkamer et al 2015 J Neurophysiology). To correctly identify whether phase-dependent modulation occurs in a muscle during rhythmic muscle activity, more than two phases would be required. Of particular importance is that the peak iEMG for ECR in this study is at the point where the authors have defined as a phase transition. If this point was defined differently, for example in relation to the shoulder, this iEMG would be phase-dependent (based on two phases only). Furthermore, if additional phases were added, such that there were 4 or more phases, there would certainly be phase-dependent modulation of TB activity. In addition to tempering this conclusion, a comment to acknowledge the limitations associated with only having only two phases in a task where there are certainly more phases would be helpful in the limitations section.

Lines 212-213: Please indicate if data were checked for normality

Figure 2: I believe the y-axes may be incorrect. Should they actually be %MVC? I thought they were normalized to maximum EMG that was determined during the sprint?

Additional comments

This is a revised version of a manuscript that I have previously reviewed for another journal, and I would like to acknowledge the improvements that have been made by the authors.

Line 117: guidelines instead of guideline?

Line 188: where should this parenthesis close?

Line 191: perhaps this could read: "which was then used to normalize the amplitude of all sub-maximal cycling trials"

Line 330-331: "They suggested the increased EMG amplitude as cycling intensity increased was due to the recruitment of additional type 2 muscle fibres..." - This seems irrelevant here. There are many reasons why EMG would increase and it does not add to the current discussion. In addition, the previous authors were speculating, so I am not sure that referencing speculation here is helpful.

Lines 409-411: Perhaps this statement could use a reference - Pearcey, Power and Button 2014?

·

Basic reporting

No comment

Experimental design

No comment

Validity of the findings

No comment

Additional comments

You have considered how electrical activity varies according to a wide range of external power outputs, as well as two distinct (push-pull) phases of the crank cycle. Data is original and adds to existing knowledge, but I have identified a few methodological issues that require some consideration and acknowledgement within the manuscript. You also identified that this exercise mode is relevant in the context of (clinical) rehabilitation; can authors, therefore, please consider/ identify how their findings may lead to practical applications.

---

## Round 0.2 · accepted · Accept

· Academic Editor

Accept

Thank you for your detailed consideration of our two reviewers' suggestions. We are ready to go ahead.

·

Basic reporting

The author's have addressed my concerns. I believe the manuscript has been much improved.

Experimental design

n/a

Validity of the findings

n/a